# Determining the burden of missed opportunities for vaccination among children admitted in healthcare facilities in India: a cross-sectional study

Nicholas Albaugh,[1] Joseph Mathew,[2] Richa Choudhary,[3] Sadasivan Sitaraman,[3] Anjali Tomar,[3] Ishumeet Kaur Bajwa,[2] Baldeep Dhaliwal,[4] Anita Shet [4]

¹Department of Epidemiology, Johns Hopkins University Bloomberg School of Public Health, Baltimore, Maryland, USA
²Pediatrics, Post Graduate Institute of Medical Education and Research, Chandigarh, Chandigarh, India
³Pediatrics, Sawai Man Singh Medical College and Hospital, Jaipur, Rajasthan, India
⁴International Health, Johns Hopkins University Bloomberg School of Public Health, Baltimore, Maryland, USA

**Correspondence to**
Dr Anita Shet; ashet1@jhu.edu

## ABSTRACT

**Objectives** Children accessing healthcare systems represent a vulnerable population with risk factors for poor health outcomes, including vaccine-preventable diseases. We aimed to quantify missed vaccination opportunities among hospitalised children in India, and identify vaccination barriers perceived by caregivers and healthcare providers.

**Design** Cross-sectional study.

**Setting** Two public-sector tertiary-care hospitals in northern India, during November 2018 and March 2019.

**Participants** We tracked 263 hospitalised children aged 1–59 months through hospital discharge, to assess vaccination status, and document catch-up vaccinations given during the hospital stay. We interviewed caregivers and healthcare providers to assess their perceptions on vaccination.

**Outcomes** Proportion of hospitalised children considered under-vaccinated for their age; proportion of missed opportunities for vaccination among under-vaccinated children who were eligible for vaccination; and vaccine coverage by antigen.

**Results** We found that 65.4% (172/263) of hospitalised children were under-vaccinated for their age when they presented to the hospital. Among under-vaccinated children, 61.0% were less than 4 months old, and 55.6% reported prior contact with a health facility for a sick visit. The proportion of under-vaccinated children in hospitals were higher compared with the general population as indicated by regional vaccination coverage data. Among under-vaccinated children who were tracked till discharge, 98.1% (158/161) remained incompletely vaccinated at discharge and were considered 'missed opportunities for vaccination'. Perceived vaccination contraindications that are not part of established contraindications included in national and international guidelines was the most common reason for healthcare providers not to vaccinate children during hospital stay. Among caregivers of under-vaccinated children, 90.1% reported being comfortable having their children vaccinated while they were sick, if recommended by the healthcare provider.

**Conclusion** This pilot study confirmed that hospitalised sick children had substantial missed vaccination opportunities. Addressing these opportunities through concerted actions involving caregivers, healthcare providers and healthcare systems can improve overall vaccination coverage.

### Strengths and limitations of this study

► This study focuses on children accessing healthcare facilities, and findings indicate that vaccination screening and catch-up vaccination at the inpatient health facility level provide an important opportunity to improve vaccination coverage and protection against vaccine-preventable diseases.

► The proportion of under-vaccinated children was higher among hospitalised children at the two regions in India compared with those in the general population as indicated by regional vaccination coverage information.

► A high proportion of hospitalised children were missed for catch-up vaccination during their admission and hospital course despite the available opportunities to provide catch-up vaccination.

► Over half of the under-vaccinated children reported prior healthcare facility contact for a sick visit, indicating that these children were well within reach of the healthcare system.

## INTRODUCTION

Although global vaccination rates have remained high over the last 10 years,[1] more than one out of every three children were not fully vaccinated in India in 2014.[2] According to the 2015–2016 National Family Health Survey (NFHS-4), vaccination rates have improved steadily with each required vaccine coverage increasing between 8 and 23 percentage points in overall vaccination coverage in India's National Immunisation Schedule over the past 10 years.[3–5] A massive nationwide effort called the Mission Indradhanush strategy was launched by the Government of India in 2014, which vaccinated approximately 25.5 million vulnerable children in a span of 2 years.[6] Nonetheless, a joint report in 2016 conducted by the WHO, United Nations and Government of India Ministry of Health and Family Welfare concluded that estimated vaccination rates tended to be overestimated

**Table 1** Study definitions for vaccination status by age range

| Age range (months) | Birth dose | First dose | Second dose | Third dose | Booster dose |
|---|---|---|---|---|---|
| <1 month | BCG | | | | |
| 1 month (4–7 weeks) | BCG OPV | OPV or f-IPV; Penta or DPT+Hep B | | | |
| 2 months (8–11 weeks) | BCG OPV | OPV or f-IPV; Penta or DPT+Hep B | OPV or f-IPV; Penta or DPT+Hep B | | |
| 3–8 months | BCG OPV | OPV or f-IPV; Penta or DPT+Hep B | OPV or f-IPV; Penta or DPT+Hep B | OPV or f-IPV; Penta or DPT+Hep B | |
| 9–15 months | BCG OPV | OPV or f-IPV; Penta or DPT+Hep B; Measles or MMR or MR | OPV or f-IPV; Penta or DPT+Hep B | OPV or f-IPV; Penta or DPT+Hep B | |
| 16–59 months | BCG OPV | OPV or f-IPV; Penta or DPT+Hep B; Measles or MMR or MR | OPV or f-IPV; Penta or DPT+Hep B; Measles or MMR or MR | OPV or f-IPV; Penta or DPT+Hep B | OPV DPT |

| Vaccination status | Definition |
|---|---|
| Fully vaccinated | Child has had all of the recommended vaccines for his/her age group as defined above |
| Partially vaccinated | Child has had at least one, but not all, of the recommended vaccines for his/her age group as defined above |
| Unvaccinated | Child has had none of the recommended vaccines for his/her age group as defined above |

BCG, Bacille Calmette-Guérin vaccine; DPT, diphtheria–tetanus–pertussis vaccine; f-IPV, fractional dose of inactivated poliovirus vaccine; Hep B, hepatitis B vaccine; MMR, measles–mumps–rubella vaccine; MR, measles–rubella vaccine; OPV, oral poliovirus vaccine; Penta, diphtheria–tetanus–pertussis–hepatitis B–Haemophilus influenzae type b pentavalent vaccine.

by administrative vaccination coverage reports and must still be improved to meet the WHO Sustainable Development Goal #3 of reducing child mortality.[7] The report acknowledged the challenges faced by the government in identifying and addressing 'missed vaccination' opportunities for children.[7] WHO defines a missed opportunity for vaccination (MOV) as any contact with health services by a child or adult who is eligible for vaccination, which does not result in the individual receiving all the vaccine doses for which he or she is eligible.[8]

Hospitalised children represent a vulnerable population with risk factors for poor health outcomes, who are more likely than the general population to be eligible for catch-up vaccinations.[9] Children admitted to healthcare facilities may be at greater risk for infectious diseases, especially vaccine-preventable illnesses.[10] These children can benefit if vaccination opportunities are identified and addressed while the child is in the hospital. An analysis using Demographic Health Surveys and Multiple Indicator Cluster Surveys estimated that if children in contact with health services for acute care were to receive their due vaccines, the potential increase in DTP3 vaccination coverage would be 3%–14% in low and middle income country (LMIC) settings, including a 12% increase in India.[10] Our study aimed to document vaccination status of hospitalised children by obtaining vaccination cards where available, verifying the medical records at hospital admission and discharge and assessing vaccination barriers perceived by caregivers and healthcare providers. Our findings will be useful for estimating the proportion of missed opportunities for vaccination among hospitalised children in India, and for understanding the local barriers for vaccination that can potentially form the basis for contextually effective interventions.

## METHODS

This study was a cross-sectional observational research study conducted at two tertiary-care government hospitals in Chandigarh and Jaipur, between November 2018 and March 2019. Children aged 1–59 months were screened for study eligibility at admission to the hospitals. Those children admitted to the hospital, and whose parents agreed to provide vaccination information and consented to the study, were enrolled. Children who had health conditions that precluded the use of vaccines, such as a known primary or secondary immune deficiency, malignancy requiring chemotherapy and children with previous reported allergies or contraindications to any vaccine or vaccine component, were excluded from enrollment. Following written informed consent from the caregiver, concerted efforts were made to obtain the vaccination card from parents during the hospital stay. If the card was not present, the parent or caregiver was requested to bring the family member card from their home or send a picture of the card using their mobile phone. Parental recall was used to document vaccination status only if the vaccination card was unavailable.

On admission, caregivers were interviewed about demographic details, household information and prior

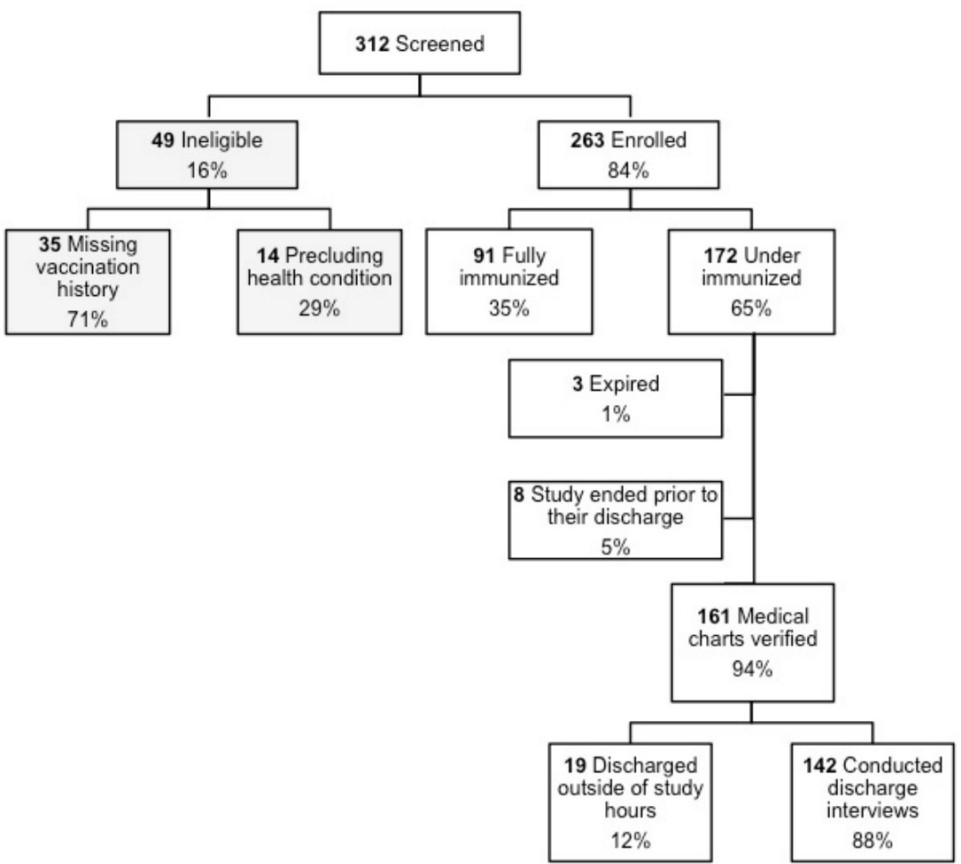

**Figure 1** Health-centre-based flowchart for determining eligibility. Missing vaccination history includes unable able to provide a vaccination card or accurately recall the vaccination history.

health-seeking behaviours. Children who had received all recommended vaccines for their age, per the prevalent National Immunisation Schedule, were classified as 'fully vaccinated'.[11] Those who had received some, but not all, of the recommended vaccines for their age group, were considered 'partially vaccinated'. Those who had received none of the recommended vaccines for their age were labelled 'unvaccinated'. Research team members used study definitions and the recommended National Immunisation Schedule to classify children who were unvaccinated or partially vaccinated as 'under-vaccinated' (table 1). This classification was verified during the analysis phase using the child's age at the time of interview and reported vaccinations. No discrepancies existed between research staff evaluation and analysis. Under-vaccinated children were tracked until discharge to record if any vaccinations, or advice regarding vaccinations, had been given during hospitalisation or at discharge. The medical chart was reviewed to document which vaccines, if any, were administered during hospitalisation. The caregivers of under-vaccinated children were interviewed at discharge to assess their perspectives on vaccination within the context of the hospital.

After the enrollment period, surveys were distributed to hospital healthcare providers involved in children's admission, treatment and discharge. Providers were asked to anonymously share their professional experiences,

vaccination practices and related policies of the hospital. Eligible healthcare providers included physicians, nurses, pharmacy personnel and postgraduate residents in the inpatient unit and the emergency unit. Questions were derived from the WHO *Planning Guide to Reduce Missed Opportunities for Vaccination* and WHO *Methodology for the Assessment of Missed Opportunities for Vaccination.*[10 12]

### Patient and public involvement

Prior to study initiation and during the development process, we piloted the study questionnaires to refine them further. We involved some caregivers of hospitalised children for this piloting purpose, and their feedback was used to re-design the data collection methods. This contribution is noted in the acknowledgement section. Patients and their caregivers were active participants in this research, but were not involved in the conduct, analysis or dissemination of this research.

### Sample size calculations

The sample size calculations were performed to enrol a total sample of 250 hospitalised eligible children and to conduct an equal number of caregiver interviews. These sample size calculations are based on the following assumptions: the estimated proportion of children admitted to the hospital who are unvaccinated or incompletely immunised is 50%; the estimated proportion of

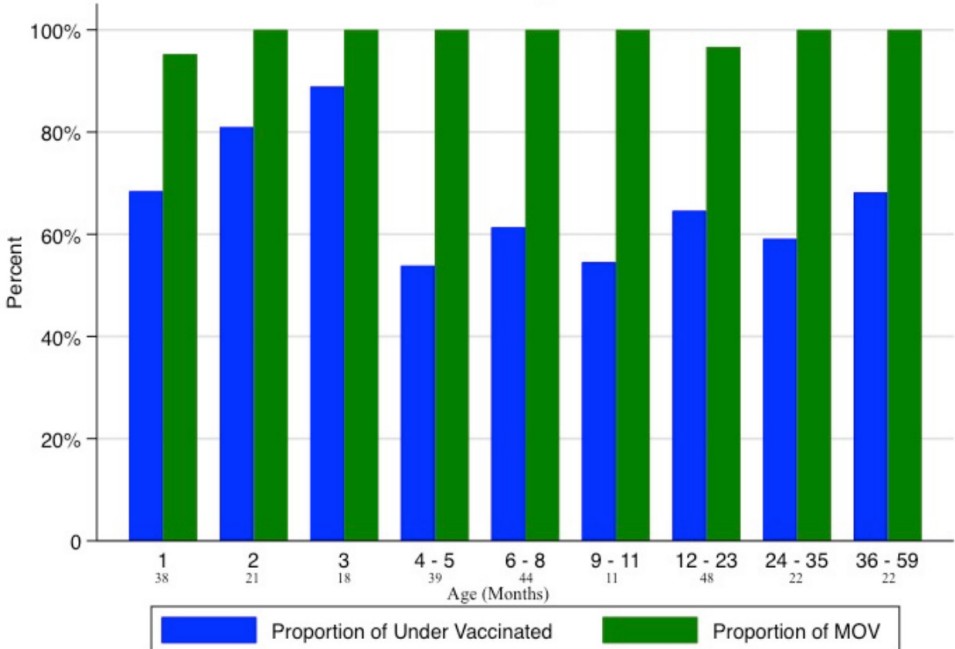

**Figure 2** The two main study outcomes are shown: proportion under-vaccinated (in blue) and proportion who missed opportunities for vaccination while in the health facility (in green) by age-range groupings (months). The number below the age-range groupings on the x-axis show the number of children within each age-range grouping. The age-range groupings correspond to ages when doses are recommended by the National Immunisation Schedule.

those eligible children who missed the opportunity to get vaccinated during their hospital visit is 30%; and the loss to follow-up of children in the study between admission and discharge (discharged before the follow-up contact, or refused/was unable to participate) is 20% (alpha 0.05, margin of error ±9%).

### Statistical analysis

Study outcomes included (1) the proportion of hospitalised children considered under-vaccinated for their age; (2) the proportion of missed opportunities for vaccination among under-vaccinated children who were eligible for vaccination[8]; and (3) vaccine coverage by antigen. In addition, we compared these estimates of vaccine coverage among our sample of hospitalised children with regional Chandigarh and Jaipur estimates from the NFHS-4 published dataset.[2] For this sub-analysis to remain comparable with the NFHS-4 datasets, we only included children aged 12–23 months old using the NFHS-4 definition of 'fully vaccinated' who have received one dose of BCG, three doses each of polio and DPT+Hib+Hep B (Penta) and one dose of measles.[13] We fitted the multivariable Poisson regression models for predictors of being under-vaccinated. Poisson regression is more appropriate than logistic regression due to the common (>10%) prevalence of being under-vaccinated and because the log binomial OR models did not converge.[14] Weight-for-age z-scores were calculated using WHO growth indicator equations.[15] Caregiver and healthcare provider survey responses were coded and analysed as binary outcomes for knowledge questions (correct/incorrect) and perception statements (agree/disagree). All statistical analyses

were conducted using Stata V.13.1 (StataCorp, College Station, Texas, USA).

### RESULTS

Of the 312 admitted children who were screened for enrollment, 49 were ineligible (35 had no vaccination card or parental recall of vaccination details and 14 had a prior precluding health condition), leaving 263 (62 at Chandigarh and 201 at Jaipur) who consented and were subsequently enrolled into the study (figure 1).

### Participant characteristics

The median age of children in our sample was 7 months (range 1–58 months), and 57.8% (152/263) were boy. First-born children made up 45.2% (119/263) of our sample. The median age of caregivers in the study was 25 years (range 18–63 years), and 78.6% (206/262) were women, with the majority of caregivers being mothers. The highest attained educational level was high school or greater among 64.5% of caregivers, and 77.2% could read and write. The primary residential setting was rural among 60.8% (160/263), with 31.1% (80/263) of children living more than 5 km away from their nearest health centre (online supplemental table S1). At the time of this study, 92.4% (243/263) of caregivers reported having a vaccination card for their children; however, only 59.7% (157/263) were able to make this available at the hospital. Parental recall was used as the immunisation source among the remaining 40.3% (106/263) (online supplemental table S1).

**Table 2** Vaccination coverage for individual vaccine antigens among hospitalised children at the study sites compared with national vaccination coverage rates from states (Chandigarh and Rajasthan NFHS-4)

| | Vaccination coverage among study population of hospitalised children (n=263) | | | Aggregated Chandigarh and Rajasthan NFHS-4 estimates (2015–2016)* |
| | Vaccinated | Age eligibility | Coverage | Coverage† |
| Vaccine dose | N | N | % | % |
|---|---|---|---|---|
| BCG | 237 | 263 | 90.1 | 88.2 |
| OPV dose 0 | 169 | 263 | 64.3 | 71.7 |
| OPV dose 1 | 182 | 263 | 69.2 | 84.8 |
| OPV dose 2 | 145 | 225 | 64.4 | 78.2 |
| OPV dose 3 | 110 | 204 | 53.9 | 63.7 |
| DPT dose 1 (DPT or Penta) | 187 | 263 | 71.1 | 83.6 |
| DPT dose 2 (DPT or Penta) | 142 | 225 | 63.1 | 78.0 |
| DPT dose 3 (DPT or Penta) | 112 | 204 | 54.9 | 69.6 |
| Measles dose 1 (measles, MR, MMR) | 55 | 69 | 79.7 | 80.4 |
| Fully vaccinated‡ | 17 | 48 | 35.4 | 62.0 |

*International Institute for Population Sciences (IIPS) and ICF. National Family Health Survey (NFHS-4) 2015–2016 (Dataset). Data Extract from IAIR72. SAV, IAKR72.SAV, IABR72.SAV, IAMR74.SAV and IAPR74.SAV. IPUMS Demographic and Health Surveys, 2018.[2]
†Coverage is from birth to 48 months of age. Age range in our sample was from 1 to 59 months.
‡NFHS defines fully vaccinated as having BCG, measles and three doses each of polio and DPT by 12–23 months of age.

## Vaccination status

Among the 263 children included in the final analysis, 34.6% (91) were fully vaccinated while 65.4% (172) were under-vaccinated. Among under-vaccinated children, 60.1% (158/263) and 5.3% (14/263) were partially vaccinated and unvaccinated, respectively. The proportion of under-vaccinated children was highest at 76.6%, (59/77) among those under 4 months of age (figure 2). The average hospital duration among those under-vaccinated was 5 days (range 1–58 days), and 47% (72/161) were classified as underweight (weight <2 z-scores below the mean age-for-weight WHO definition).[15] Prior health facility contact for a sick visit at any time since birth, and within the last month, was reported in 55.6% (93/167) and 40.7% (68/167) of under-vaccinated children, respectively.

The vaccinations with the lowest coverage among all included children were OPV3 (53.9%), DPT3 (54.9%) and DPT2 (63.1%). Our sample also had less than 80% vaccination coverage for the following vaccines: OPV0 (64.3%), OPV1 (69.2%), OPV2 (64.4%) and MCV1 (79.7%). The coverage of MCV2 (measles, MR and MMR), OPV booster dose and DPT booster dose was 40.9%, 56.8% and 59.1%, respectively (online supplemental table S2). Using the NHFS-4 study definition of full vaccination, our study sample (children aged 12–23 months old) had a proportion of 35.4% (17/48) fully vaccinated children, while the reported 2015–2016 estimated national average for full vaccination coverage was 62.0%.[13] Furthermore, our sample had lower coverage than the regional average for every vaccine, except BCG (90.1%) and MCV1 (79.7%)[2] (table 2). A similar look at the two states separately showed similar results (online supplemental table S3).

Among 161 under-vaccinated children who had a medical chart available at discharge, 98.1% (158/161) represented an MOV. The most common admission diagnoses were lower respiratory tract infection, sepsis, fever for evaluation and epilepsy. Three children who expired during the study and eight children who remained admitted in the hospital at study end (figure 1) were excluded from the analysis. There was no significant difference of missed opportunities for vaccination by age (figure 2). Two children (aged 1 month and 17 months, respectively) were given their missed vaccinations in the hospital, and a third child (aged 3 months) visited the immunisation clinic for catch-up vaccination prior to discharge. The discharge interview was completed among 88% (142/161) of caregivers of under-vaccinated children while the remaining 12% (19/161) were discharged outside of study hours. Among those interviewed, 62.9% of caregivers knew the correct purpose of vaccination ('to prevent diseases'); 8.5% reported having their child's vaccination history verified by staff; 1.4% reported receiving vaccine recommendations and information on vaccine clinic locations; 1.4% were told about the potential vaccination side effects; and 90.1% reported they were comfortable having their children vaccinated while they were sick. When asked about vaccine reminder services or text message reminders for vaccinations after discharge, 95% of caregivers said they would support such measures, and 32.9% suggested there should be more vaccination outreach services.

**Table 3** Prevalence rate ratio of being under-vaccinated based on multiple predictors (n=263)

| | Total | Under vaccinated | Univariate analysis | | | Multivariable analysis* | | |
|---|---|---|---|---|---|---|---|---|
| | N (%) | N (%) | PR | 95% CI | P value | Adj. PR | 95% CI | P value |
| **Child gender** | | | | | | | | |
| Female | 111 (42) | 71 (64) | Referent | | | Referent | | |
| Male | 152 (58) | 101 (66) | 1.03 | 0.87 to 1.24 | 0.68 | 0.97 | 0.80 to 1.18 | 0.78 |
| **Child age (months)** | | | | | | | | |
| >4 | 186 (71) | 113 (61) | Referent | | | Referent | | |
| <4 | 77 (29) | 59 (77) | 1.26 | 1.06 to 1.49 | <0.01* | 1.04 | 0.83 to 1.29 | 0.74 |
| **Child birth order** | | | | | | | | |
| First | 119 (45) | 71 (60) | Referent | | | Referent | | |
| After (second, third, fourth) | 144 (55) | 101 (70) | 1.18 | 0.98 to 1.41 | 0.08 | 1.15 | 0.93 to 1.42 | 0.19 |
| **Residency** | | | | | | | | |
| Urban/suburban | 103 (39) | 61 (59) | Referent | | | Referent | | |
| Rural | 160 (61) | 111 (69) | 1.17 | 0.97 to 1.42 | 0.1 | 1.14 | 0.91 to 1.43 | 0.26 |
| **Caregiver education (N=262)†** | | | | | | | | |
| College or beyond college | 85 (32) | 44 (26) | Referent | | | Referent | | |
| High school | 84 (32) | 57 (33) | 1.31 | 1.02 to 1.69 | 0.04 | 1.3 | 0.97 to 1.75 | 0.08 |
| Less than high school | 93 (35) | 70 (41) | 1.45 | 1.15 to 1.84 | <0.01* | 1.29 | 0.99 to 1.68 | 0.06 |
| **Health facility contact (N=257) †** | | | | | | | | |
| Over 3 months | 44 (17) | 25 (15) | Referent | | | Referent | | |
| Within last 3 months | 94 (37) | 68 (41) | 1.27 | 0.96 to 1.70 | 0.1 | 1.13 | 0.81 to 1.57 | 0.49 |
| Never | 119 (46) | 74 (44) | 1.09 | 0.82 to 1.47 | 0.55 | 1.01 | 0.73 to 1.40 | 0.94 |
| **Health facility distance (km) (N=257) †** | | | | | | | | |
| 0–5 | 177 (69) | 111 (63) | Referent | | | Referent | | |
| 5–50 | 80 (31) | 56 (70) | 1.12 | 0.93 to 1.34 | 0.24 | 0.99 | 0.80 to 1.22 | 0.92 |
| **Vaccination decision maker (N=257)†** | | | | | | | | |
| Mother involved | 214 (83) | 133 (62) | Referent | | | Referent | | |
| Mother not involved | 43 (17) | 33 (77) | 1.23 | 1.02 to 1.50 | 0.03* | 1.16 | 0.92 to 1.46 | 0.22 |
| **Had rotavirus vaccine dose 1** | | | | | | | | |
| Yes | 74 (28) | 31 (42) | Referent | | | Referent | | |
| No | 189 (72) | 141 (75) | 1.78 | 1.34 to 2.36 | <0.01* | 1.4 | 1.01 to 1.95 | 0.05 |
| **Prior vaccination setting (N=223)†** | | | | | | | | |
| Government | 202 (91) | 124 (61) | Referent | | | Referent | | |
| Private | 21 (9) | 17 (81) | 1.32 | 1.04 to 1.67 | 0.02* | 1.35 | 1.07 to 1.71 | 0.02* |
| **Have their vaccination card** | | | | | | | | |
| Present at hospital | 154 (59) | 84 (55) | Referent | | | Referent | | |
| Absent at hospital | 109 (41) | 88 (81) | 1.48 | 1.25 to 1.76 | <0.00 | 1.18 | 0.94 to 1.47 | 0.15 |

*Adjusted for all variables in the table.
†Number in parenthesis refers to denominator in each category if different from the main denominator of 263.

## Predictors of being under-vaccinated

In univariate analysis, there was a significant association between being under-vaccinated and the following determinants: children aged less than 4 months compared with older children (prevalence ratio (PR) 1.26, 95% CI 1.04 to 1.49); parents with lower than high school educational attainment (PR 1.45, 95% CI 1.15 to 1.84); children of mothers not involved in vaccination decisions (PR 1.23, 95% CI 1.02 to 1.50); those previously vaccinated in a private health facility (PR 1.32, 95% CI 1.04 to 1.67); and children whose vaccination information was collected through parental recall due to the lack of an available vaccination card (PR 1.48, 95% CI 1.25 to 1.76). In multivariable analysis, there were similar trends between the outcome of being under-vaccinated and the predictors, although there was no statistical significance (table 3).

## Healthcare providers

Among 21 healthcare providers who participated across both sites (physicians and nurses from the paediatric inpatient units, emergency unit and immunisation clinic), 57% were female, the median age was 27 years and 76.2% reported having more than 1 year of experience working in the healthcare field. Among the healthcare providers, 85.7% had incorrect knowledge of vaccination contraindications, listing low-grade fever, pneumonia or diarrhoea, and acute condition-related hospitalisation as absolute contraindications. Although 95% agreed that it was appropriate to vaccinate a hospitalised child prior to discharge, 80% had concerns about vaccinating a child recovering from an acute illness, and 80% believed vaccines were not available in the inpatient unit. Reported barriers to vaccination of hospitalised children were illness of the hospitalised child (71.4%), non-availability of immunisation staff in the inpatient ward (71.4%), non-provision of vaccines in the inpatient ward (61.9%) and the focus of hospital health providers on management of acute illness rather than primary preventative measures (47.6%). Most (94.4%) believed these barriers could be addressed. Strategies suggested for improving catch-up vaccinations were as follows: providing a reminder service at discharge for children missing vaccinations (85.7%), improving communication regarding vaccination status screening (76.2%) and improving vaccine delivery logistics and communication among hospital staff (66.7%).

## DISCUSSION

We observed a high proportion of under-vaccinated children among those hospitalised. This proportion was substantially higher than the national and regional averages for the period of 2015–2016.[2] The overall proportion of missed opportunities for vaccination during hospitalisation was 98.1%, which was significantly higher in comparison with other observational studies that included inpatient and outpatient healthcare facilities (25%–43%).[9 16–19] A recent meta-analysis from seven LMICs in Asia, Africa and the Americas found a pooled prevalence of missed vaccination opportunities among children accessing healthcare facilities at 32% over a 22-year period.[20] An observational survey study in Mumbai found that healthcare providers missed opportunities to vaccinate children 80% of the time during a visit for an illness compared with only 0.7% of the time during a visit for a healthy child.[21] Using WHO standardised evaluation instruments, a study across 30 health facilities in Chad and Malawi found a higher proportion of missed vaccination opportunities in non-vaccination purpose visits (77%–92%), as seen in our study, compared with vaccination visits (47%–51%).[22] A descriptive analysis of private and public health facilities in four African countries found that 93%–99% of eligible children remained under-vaccinated during outpatient non-vaccination, acute-care visits based on the provider's response.[23] In line with these studies, our findings suggest that reducing missed vaccination opportunities can be impactful during inpatient non-vaccination hospital visits.

We observed that vaccination coverage among our sample of hospitalised children was substantially lower than nationally reported and regional vaccination coverage estimates for infant vaccination in general. Hospitalised children accessing healthcare in the public sector, such as our study sites, usually come from lower income quintiles from outside the cities seeking tertiary care and are often the most vulnerable to repeated events of ill health.[24] Individuals in the lowest income quintiles are particularly vulnerable to illness as they are at higher risk of being malnourished, living in crowded conditions and having less access to clean water.[25] In addition to conditions that contribute to poor health outcomes, the poorest children in India are less likely to be fully vaccinated, leaving them even more susceptible to preventable diseases, and leaving their caregivers susceptible to catastrophic health costs. We found that children who were in contact with a health facility in the last 3 months were less likely to be fully vaccinated, suggesting a continual pattern of missed opportunities for vaccination. It is important in chracterising this association to understanding that parents may not routinely seek vaccines or be aware that their children are under-vaccinated, hence it is crucial for facilities to devote more attention to vaccination screening among admitted children. Addressing missed opportunities for vaccination is even more important considering the COVID-19 pandemic, and its negative impact on routine vaccine-seeking behaviour.[26] Interventions and communication strategies will be particularly important, as this will facilitate easy catch-up for vaccines missed during the pandemic, and it will also ensure there is infrastructure in place to rollout an eventual COVID-19 vaccine.

In this study, some healthcare provider perceptions could be significant barriers to provision of vaccinations. A study in Chad and Malawi found that 92% and 88% of healthcare providers, respectively, were unable to correctly identify valid contraindications for vaccination.[22] Our observations on a limited number of healthcare provider perceptions indicate that interventions targeting vaccination catch-up should focus on facility barriers training on vaccination schedules and contraindications; encourage screening for childhood vaccination status on admission; and ensure that staff at inpatient and sick visit areas as well as in immunisation clinics are well informed of the need for routine and catch-up childhood vaccination services. In studies that investigated the determinants of vaccination, barriers have been generalised into categories that included health system barriers (cost, infrastructure); provider barriers (poor knowledge of contraindications, poor access to immunisation records); and caregiver barriers (economic problems, poor understanding of side effects).[27 28] As barriers to vaccination are unique across regions, with varying determinants, understanding them at the caregiver and provider level can lead to the design of context-specific interventions to drive vaccine uptake in the inpatient setting. A systematic review of barriers to

vaccination and missed opportunities found risk factors associated with being under-vaccinated which were similar to our findings.[20] This review additionally identified lower household socioeconomic status, late initiation of the child's vaccination, parental information gap, provider practices and a variety of clinic factors as contributing factors.[28] The first 3 months of life is an intense period for vaccinations, and is also a vulnerable time for illnesses. More research is needed to further explore the association among these risk factors, as well as test interventions that bring together community and facility-based care to ensure complete vaccination of all children. The MOV strategy designed by the WHO describes the standardised global practice for reducing missed opportunities for vaccination, which includes implementing the interventions, providing supportive supervision and monitoring progress, conducting rapid field evaluations of outcome and incorporating into the long-term vaccination plans to ensure sustainable goals.[29] Implementing screening and communication strategies to reduce missed opportunities for vaccination at the facility level will be crucial and cost-effective for enhancing and optimising vaccination coverage.

This study was limited by a small sample size and was completed within a short duration. We included those without a vaccination card and depended on parental recall in some children, which may have affected reliability.[30] A systematic review on validity of parental recall for childhood vaccination surveys called into question the reliability of this method, and also concluded that excluding caregiver recall could result in considerable bias in vaccine coverage estimations.[31] We took considerable precautions to ensure that hospital healthcare providers did not become greatly aware of the study details, and this minimised influencing their behaviour during the study.[32] Caregiver perspectives were assessed only among those whose children were under-vaccinated, due to limitations in follow-ups. Admission diagnoses were not routinely captured for the purpose of the study and hence are not reported in the analyses. Healthcare provider responses were not analysed based on professional job title or years of experience, as the numbers were too small to show significant differences. This study was only conducted at two government hospitals, which could limit the generalisability of the results. In India, patients who receive medical care at government-run hospitals have lower out-of-pocket expenses compared with those accessing healthcare in the private sector, potentially biassing our sample based on facility selection.[24] While the two hospitals are in different states, there was no private hospital comparison in this preliminary study. However, we believe that focusing on government-run hospitals is a strength of our work, as it enabled us to characterise a population in India that is more prone to being under-vaccinated, while also allowing us to capture missed opportunities for vaccination among vulnerable populations.[24 28] Moreover, the public sector is responsible for providing 60% of antenatal visits, as well as 90% of vaccination doses to children

in India.[24] By conducting this research in the public sector, we were able to gain an understanding of missed opportunities, as well as barriers to vaccinations, in the setting where the vast majority of children are vaccinated. Interventions based on these insights have the potential to have a profound impact on vaccine uptake in India. This pilot study helped establish feasibility assessments and power calculations for an interventional trial for addressing missed opportunities for vaccination that has been initiated in India and Nigeria.

## CONCLUSION

In conclusion, we found a high proportion of under-vaccination among hospitalised children, and the proportion of missed vaccination opportunities was high in these healthcare facilities. Our observation that a substantial proportion of under-vaccinated children have had prior contact with healthcare facilities underscores the need for interventions that repurpose existing resources at these healthcare facilities to ensure that vaccination opportunities are not missed. Addressing the system for vaccination screening at healthcare facilities, improving communication strategies targeted at both caregivers and healthcare providers and removing logistic barriers to vaccine access within the healthcare facility can significantly improve vaccination inequities.

**Acknowledgements**  The authors thank Laura Lochlainn, Ikechukwu Udo and Stephanie Shendale of the Department of Immunisation, Vaccines & Biologicals, WHO headquarters (WHO-HQ) for early discussion and input on the study, and they thank Farhad Khan for helpful discussions on the analysis and drafting of the manuscript. The authors are also grateful to the caregivers of hospitalised children who assisted in refining and piloting the study questionnaires prior to study initiation.

**Contributors**  NA, JM and AS conceived the study with contributions from SS. NA prepared the dataset and conducted the analysis of the data. NA, BD and AS prepared the first draft of the manuscript. NA, RC, AT and IKB were responsible for overseeing the acquisition and management of the data for the study. All authors reviewed the drafts of this manuscript, provided critical input and approved the final version for submission.

**Funding**  This MOVAHC study was funded through the Bill and Melinda Gates Foundation Gates Grand Challenges (No. OPP1217304).

**Competing interests**  None declared.

**Patient consent for publication**  Not required.

**Ethics approval**  This study was approved by the Institutional Review Board at the Johns Hopkins Bloomberg School of Public Health and local Institutional Ethics Committees at the Postgraduate Institute of Medical Education and Research, Chandigarh and Sawai Man Singh Medical College and attached hospitals, Jaipur. Written informed consent was obtained from participants prior to enrollment and implementation of study procedures. All anonymised data were collected on paper and entered into REDCap.

**Provenance and peer review**  Not commissioned; externally peer reviewed.

**Data availability statement**  Data are available upon reasonable request. De-identified data generated by this research will be made available after publication of this article upon request to the corresponding author.

of the translations (including but not limited to local regulations, clinical guidelines, terminology, drug names and drug dosages), and is not responsible for any error and/or omissions arising from translation and adaptation or otherwise.

**ORCID iD**
Anita Shet http://orcid.org/0000-0002-7204-8164

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
