## [Reviewer comments · BMJ Open]

ARTICLE DETAILS

TITLE (PROVISIONAL)	Determining the burden of missed opportunities for vaccination among children admitted in healthcare facilities in India: a cross-sectional study
AUTHORS	Albaugh, Nicholas; Mathew, Joseph; Choudhary, Richa; Sitaraman, Sadasivan; Tomar, Anjali; Bajwa, Ishumeet; Dhaliwal, Baldeep; Shet, Anita

VERSION 1 – REVIEW

REVIEWER	Tarun Shankar Choudhary Society for Applied Studies, India
REVIEW RETURNED	20-Dec-2020

GENERAL COMMENTS	This is a well written article. Some minor suggestions are listed below: 1. It will be useful to present the indication for admission in the study participants as the readers will get an idea of the seriousness of the underlying condition2. The authors have stated that information was based on parental recall in some children, but 40% of the sample will not constitute "some" children. Presenting an analysis for only those with vaccination card based data (as a supplementary table) will help the readers.3. Although the authors have referred to the study as pilot study it would be useful to specify if any sample size calculation was done or were all children reporting to these facilities eligible for participation in the study?4. Qualitative research methods would have been better for understanding the perspectives of care givers and healthcare providers.5. It should be clarified if the consent was verbal or written.6. Typographical error in line 319
--

REVIEWER	Luke Shenton Drexel University College of Medicine USA
REVIEW RETURNED	04-Jan-2021

GENERAL COMMENTS	Introduction: Overall well written and provides clear context and defines a clear objective. Line 61: what does 8-23% represent? How changes differed between states, a confidence interval, different survey sources? Please make less ambiguous Line 77: "diseases, especially" remove the and
---

	Line 84: Please cite a few of these studies that have attempted to quantify missed opportunities Methods: Overall methods were clear and reproducible. Line 101: enrollment Line 120: crossed out? should this be there or not? Line 121: What was the rationale for only interviewing caregivers of under vaccinated children as opposed to interviewing everyone and see how perspectives differed based on immunization status of the child? Line 130: could consider supplementary materials showing the survey used A justification for analyzing the Jaipur and the Chandigarh together would be helpful. I was left wondering whether the data was homogenous between centers or if large disparities existed. I'm not sure if much can be gained from comparing the data of just 2 different centers added together to overall national rates. Comparing data separately, to respective state rates from NFHS-4 could be interesting. Results: Line 201: While interesting to compare data to national levels, I'd be more interested in seeing how the data from your two centers compared to the states they were in. See comment in methods. Line 213: Was any attempt made to contact them for completion of survey? Line 215: Were any caregivers probed deeper? For instance of those who didn't know the correct purpose of vaccinations, did they say what they thought the correct purpose was? Did those who were not comfortable having their sick children vaccinated say why? Line 231: Any effort made to estimate accuracy of parental recall. I wonder if the different in vaccination was real or due to inaccuracy of parental recall? Line 238: May be beyond scope and I understand if this can't be addressed within this manuscript. However, I would be interested in different in thoughts of providers based on nurse vs MD or based on unit they work in, etc. Line 245: Are these barriers all perceived by staff or are any of them true barriers? Some of these comments may be better answered in the discussion than results section. Discussion: Line 259: You're comparing your study (inpatient) to studies looking at in and out patient facilities. Are there any other studies that also just look at inpatient visits that you could cite as a comparison? I'm just speculating but could part of the different you mention here be because vaccination at outpatient visits is more likely seeing as the acute concern is probably less pressing?
--	---

VERSION 1 – AUTHOR RESPONSE

Reviewer: 1

Dr. Tarun Choudray, Society for Applied Studies, New Delhi

Comments to the Author:

This is a well written article.

Some minor suggestions are listed below:

1. It will be useful to present the indication for admission in the study participants as the readers will get an idea of the seriousness of the underlying condition

Response:

We agree with the reviewer that the admission diagnosis would have been useful information. Unfortunately, we did not capture this information clearly in the case report forms. In addition, at one of the sites (Chandigarh) we worked in the respiratory ward, and hence lung-related disorders were the most common. In general, we note that the most common admission diagnoses were, lower respiratory tract infection, sepsis, fever for evaluation, and epilepsy. We have added this sentence in the main document in page/line no: 11 /238. The important aspect to note is that these were all children with conditions serious enough to warrant hospital admission, and this is target population we were attempting to characterize with regard to immunization status.

2. The authors have stated that information was based on parental recall in some children, but 40% of the sample will not constitute "some" children. Presenting an analysis for only those with vaccination card based data (as a supplementary table) will help the readers.

Response:

We agree with the reviewer that a substantial number of children (about 40%) did not have their vaccine card with them. In univariate analysis presented in Table 3, we show that among children with vaccine cards, 55% (84/154) are under-vaccinated, while among those without vaccine cards, 81% (88/109) are under-vaccinated ($p = <0.00$). Clearly, there was a significant association between being under-vaccinated and children whose vaccination information was collected through parental recall due to the lack of an available vaccination card. This is noted in the Results section as well. A supplementary table with prevalence rate ratio of being under-vaccinated and multiple predictors (such as Table 3) for only vaccine card-carrying children would result in very small numbers, and may not add any additional inference.

3. Although the authors have referred to the study as pilot study it would be useful to specify if any sample size calculation was done or were all children reporting to these facilities eligible for participation in the study?

Response:

We appreciate the reviewer's point and wanted to clarify the sample size calculations were conducted to enroll a total sample of 250 hospitalized eligible children and conduct an equal number of caregiver interviews. These sample size calculations are based on the following assumptions: the estimated proportion of children admitted to the hospital who are unvaccinated or incompletely immunized is 50%, the estimated proportion of those eligible children who missed the opportunity to get vaccinated during their hospital visit is 30%, and the loss to follow up of children in the study between admission and discharge (discharged before the follow-up contact; or refused/was unable to participate, etc) is 20% (alpha of 0.05, margin of error of $\pm 9\%$). The pilot study helped establish power

calculations required for the ongoing larger trial on missed opportunities for vaccination equity (MOVE).

4. Qualitative research methods would have been better for understanding the perspectives of care givers and healthcare providers.

Response:

We thank the reviewer for the suggestion. Although we were not able to conduct qualitative research methods in this pilot test, our interventional trial (MOVE) for addressing missed opportunities for vaccination includes qualitative research methods such as in-depth interviews with parents, and co-creation workshops with healthcare providers to understand barriers and solutions for missed opportunities for vaccination.

5. It should be clarified if the consent was verbal or written.

Response:

We appreciate the reviewer's suggestion. We have clarified in the ethical considerations subsection (line 165) that written informed consent was taken from participants prior to enrollment and implementation of study procedures.

6. Typographical error in line 319

Response:

We have corrected the spelling of "monitoring" in the line.

Reviewer: 2

Mr. Luke Shenton, University of Michigan School of Public Health

Comments to the Author:

Introduction:

Overall well written and provides clear context and defines a clear objective.

Line 61: what does 8-23% represent? How changes differed between states, a confidence interval, different survey sources? Please make less ambiguous

Response:

We thank the reviewer for the comment. We have revised the line to clarify that the range of 8-23 percentage points represents each required vaccine coverage increasing between 8-23 percentage points in overall vaccination coverage in India's National Immunization Schedule over the past ten years.

Line 77: "diseases, especially" remove the and

Response:

We have removed the word "and".

Line 84: Please cite a few of these studies that have attempted to quantify missed opportunities

Response:

In the Discussion section, we have cited and described several studies (citations #16-23) that have attempted to quantify missed opportunities. Hence, we have removed the sentence from the Introduction section that mentions previous studies to prevent redundancy or repetition.

Methods:

Overall methods were clear and reproducible.

Line 101: enrollment

Response:

We have corrected the spelling of “enrollment” in the line.

Line 120: crossed out? should this be there or not?

Response:

We have removed the sentence.

Line 121: What was the rationale for only interviewing caregivers of under vaccinated children as opposed to interviewing everyone and see how perspectives differed based on immunization status of the child?

Response:

We appreciate the Reviewer’s comments in this regard. The main rationale for only interviewing caregivers of under vaccinated children as opposed to interviewing everyone was our study only followed under vaccinated children through to discharge. Given the limited staff on the study, we were limited in our participant follow up rate. This limitation prevented us from learning how caregiver perspectives differed based on immunization status of the child. We have included this in the limitation section of the manuscript. This limitation is being addressed in the current MOVE trial, where all caregivers are followed-up through discharge and their perspectives are being documented.

Line 130: could consider supplementary materials showing the survey used

Response:

We thank the reviewer for the suggestion. We have added our two questionnaires for caregivers and one for healthcare workers as supplementary material.

A justification for analyzing the Jaipur and the Chandigarh together would be helpful. I was left wondering whether the data was homogenous between centers or if large disparities existed. I'm not sure if much can be gained from comparing the data of just 2 different centers added together to overall national rates. Comparing data separately, to respective state rates from NFHS-4 could be interesting.

Response:

We appreciate the reviewer’s point and want to clarify we did analyze the data in Jaipur and Chandigarh separately at our initial analyses. These analyses (seen in Supplemental Table 3). showed there were no major differences at the sites and the separated numbers were too small to report for each variable. We were aiming to show our population of hospitalized children as a whole compared to reported national routine vaccination rather than showing geographical differences.

Results:

Line 201: While interesting to compare data to national levels, I'd be more interested in seeing how the data from your two centers compared to the states they were in. See comment in methods.

Response:

We appreciate the reviewer's suggestion. As previously mentioned, reported NFHS-4 coverages for the state of Rajasthan (Jaipur) and the city of Chandigarh were not significantly different from each other or the national average.

Line 213: Was any attempt made to contact them for completion of survey?

Response:

We thank the reviewer for their comment and wanted to clarify that following patients after discharge from the hospital was beyond the scope of the study, as we did not have specific staff to make repeated follow-up calls or make home visits.

Line 215: Were any caregivers probed deeper? For instance of those who didn't know the correct purpose of vaccinations, did they say what they thought the correct purpose was? Did those who were not comfortable having their sick children vaccinated say why?

Response:

We appreciate the reviewer's suggestion. Caregivers were only asked "Could you tell me the purpose of vaccines?". Their responses were matched to either correct answers, incorrect answers, or reported not knowing the answer. They were not probed deeper after their answer if they did not know the purpose of vaccination. There was no follow up question to caregivers who were comfortable having their sick children vaccinated.

Line 231: Any effort made to estimate accuracy of parental recall. I wonder if the different in vaccination was real or due to inaccuracy of parental recall?

Response:

Since we did not have a way of cross-checking with the vaccine cards, we did not have a way assessing the accuracy of parental recall. If the card could not be presented before discharge, the parent was asked if they could recall their child's immunization history. Parents were prompted to recall the site of injection and the age of the child at the time of the infection, to validate that the information is reliable before enrolling. Several individual studies and systematic reviews call into question the reliability of parental recall, but also conclude that excluding caregiver recall could result in considerable bias in vaccine coverage estimates.

Line 238: May be beyond scope and I understand if this can't be addressed within this manuscript. However, I would be interested in different in thoughts of providers based on nurse vs MD or based on unit they work in, etc.

Response:

We thank the reviewer for pointing out this important limitation; unfortunately, we had such a small sample size that we were unable to report based on difference of job title. In our sample analysis, we did note there was a difference in attitude based on years of experience with those with more experience were more likely to know the correct contraindications for vaccination. However, because we had such a small sample size, we did not feel comfortable reporting this and giving a false generalization.

Line 245: Are these barriers all perceived by staff or are any of them true barriers?

Response:

We believe that any 'perceived' barrier is a 'true' barrier that needs to be addressed accordingly. For example, barriers due to non-awareness or pre-conceived notions are as real and need solutions in the same manner as logistic barriers such as vaccine stockouts or clinic timings. Due to the fact the answers were recorded through anonymous survey, we felt more comfortable reporting them as 'perceived barriers' since that was the scope of this pilot testing. This study was specifically conducted to inform the design of a larger interventional trial for addressing missed opportunities for vaccination.

Some of these comments may be better answered in the discussion than results section.

Discussion:

Line 259: You're comparing your study (inpatient) to studies looking at in and out patient facilities. Are there any other studies that also just look at inpatient visits that you could cite as a comparison? I'm just speculating but could part of the difference you mention here be because vaccination at outpatient visits is more likely seeing as the acute concern is probably less pressing?

Response:

It is certainly possible that outpatient visits may be seen as less pressing, and there is a greater opportunity to check for and address missed vaccinations. It is also possible that the vaccination clinic is generally more accessible in an outpatient setting than in an inpatient setting. In our literature search, we found that a majority of the studies in this area were conducted in outpatient settings, or at best, mixed settings, and that there were hardly any large studies that only looked at inpatient visits. This was one of the main rationales for conducting this study as we felt it was important to document the variations in findings and barriers in solely inpatient settings.

We attempt to follow up in the paragraph (lines 294-300) with studies that looked at missed opportunities for vaccination of visits for an illness and non-vaccination purpose visits to try to make a bridge in our comparison.